# Distributional Reinforcement Learning for Risk-Sensitive Policies

**Shiau Hong Lim**
IBM Research, Singapore
shonglim@sg.ibm.com

**Ilyas Malik**
IBM Research, Singapore
malikilyas1996@gmail.com

## Abstract

We address the problem of learning a risk-sensitive policy based on the CVaR risk measure using distributional reinforcement learning. In particular, we show that the standard action-selection strategy when applying the distributional Bellman optimality operator can result in convergence to neither the dynamic, Markovian CVaR nor the static, non-Markovian CVaR. We propose modifications to the existing algorithms that include a new distributional Bellman operator and show that the proposed strategy greatly expands the utility of distributional RL in learning and representing CVaR-optimized policies. Our proposed approach is a simple extension of standard distributional RL algorithms and can therefore take advantage of many of the recent advances in deep RL. On both synthetic and real data, we empirically show that our proposed algorithm is able to learn better CVaR-optimized policies.

## 1 Introduction

In standard reinforcement learning (RL) (Sutton and Barto, 2018), one seeks to learn a policy that maximizes an objective, usually the expected total discounted rewards or the long-term average rewards. In stochastic domains, especially when the level of uncertainty involved is high, maximizing the expectation may not be the most desirable since the solution may have high variance and occasionally performs badly. In such scenarios one may choose to learn a policy that is more risk-averse and avoids bad outcomes, even though the long-term average performance is slightly lower than the optimal.

In this work we consider optimizing the conditional value-at-risk (CVaR) (Rockafellar and Uryasev, 2000), a popular risk measure, widely used in financial applications, and is increasingly being used in RL. The CVaR objective focuses on the lower tail of the return and is therefore more sensitive to rare but catastrophic outcomes. Various settings and RL approaches have been proposed to solve this problem (Petrik and Subramanian, 2012; Chow and Ghavamzadeh, 2014; Chow and Pavone, 2014; Tamar et al., 2015; Tamar et al., 2017; Huang and Haskell, 2020). Most of the proposed approaches, however, involve more complicated algorithms than standard RL algorithms such as Q-learning (Watkins and Dayan, 1992) and its deep variants, e.g. DQN (Mnih et al., 2015).

Recently, the distributional approach to RL (Bellemare et al., 2017; Morimura et al., 2010) has received increased attention due to its ability to learn better policies than the standard approaches in many challenging tasks (Dabney et al., 2018a,b; Yang et al., 2019). Instead of learning a value function that provides the expected return of each state-action pair, the distributional approach learns the entire return distribution of each state-action pair. The approach itself is a simple extension to standard RL and is therefore easy to implement and able to leverage many of the advances in deep RL.

Since the entire distribution is available, one naturally considers exploiting this information to optimize for an objective other than the expectation. Dabney et al. (2018a) presented a simple way to

do so for a family of risk measures including the CVaR. The theoretical properties of such approach, however, are not clear. In particular, it is not clear whether the algorithm converges to any particular variant of CVaR-optimal policy. We address this issue in this work.

Our main contribution is to first show that the commonly-used action-selection strategy for CVaR in distributional RL, proposed by (Dabney et al., 2018a) and (Keramati et al., 2020), among others, converges to neither the dynamic, Markovian CVaR nor the static CVaR even if the optimal CVaR policy is stationary and Markov. Secondly, we propose a new distributional Bellman operator and show that for the class of stationary CVaR-optimal policies, the optimal value distribution is a fixed point, and one can extract this optimal policy through distributional RL. Empirically, we show that the proposed approach learns better policies in terms of the CVaR objective on both synthetic and real-world problems.

We close the introduction section with some references to related works. We formally present our problem setup as well as our main analytical results in Section 2. Section 3 describes our proposed algorithm while Section 4 presents our empirical results. Finally, we conclude with a discussion on future works.

## 1.1 Related Works

The literature on distributional RL has been greatly expanded recently (Morimura et al., 2010; Bellemare et al., 2017; Barth-Maron et al., 2018; Dabney et al., 2018a,b; Yang et al., 2019). Most of these works focus on the modeling aspects, such as the choice of representations for the value distributions. The approach has been used to enhance exploration in RL (Mavrin et al., 2019) and in risk-sensitive applications (Bernhard et al., 2019).

Solving Markov decision processes (MDP) with risk-sensitive objectives have been addressed in many works (Howard and Matheson, 1972; Ruszczynski, 2010; Bäuerle and Ott, 2011), including RL approaches (Borkar, 2001; Tamar et al., 2012; L.A. and Ghavamzadeh, 2013). In particular, Chow and Ghavamzadeh (2014); Tamar et al. (2015) deal with the static CVaR objectives while Petrik and Subramanian (2012); Chow and Pavone (2014) deal with the dynamic CVaR objectives. Tamar et al. (2017) proposed a policy-gradient approach that deals with both the static and the dynamic CVaR objectives. Closest to ours is the work by Stanko and Macek (2019), whose proposed approach also makes use of distributional RL but their policy evaluation step uses Markov action-selection strategy. It is not clear whether their overall approach properly optimizes either the static or the dynamic CVaR.

## 2 Problem Setup and Main Results

We consider a discrete-time MDP $\mathcal{M}$ with state space $\mathcal{X}$ and action space $\mathcal{A}$. For simplicity we assume that $\mathcal{X}$ and $\mathcal{A}$ are finite, although our results and algorithm can be readily extended to more general state-action spaces. We assume that the rewards are bounded and drawn from a countable set $\mathcal{R} \subset \mathbb{R}$. Given states $x_t, x_{t+1} \in \mathcal{X}$ for any $t \in \{0, 1, \ldots\}$, the probability of receiving reward $r_t \in \mathcal{R}$ and transitioning to $x_{t+1}$ after executing $a_t \in \mathcal{A}$ in $x_t$ is given by $p(r_t, x_{t+1}|x_t, a_t)$. Without loss of generality we assume a fixed initial state $x_0$, unless stated otherwise. Given a policy $\pi : \mathcal{H} \to \mathcal{P}(\mathcal{A})$, where $\mathcal{H}$ is the set of all histories so far $h_t := (x_0, a_0, r_0, x_1, a_1, r_1, \ldots, x_t) \in \mathcal{H}$, and $\mathcal{P}(\mathcal{A})$ the space of distributions over $\mathcal{A}$, its expected total discounted reward over time is given by

$$V^\pi := \mathbb{E}_p^\pi \left[ \sum_{t=0}^\infty \gamma^t r_t \right]$$

where $\gamma \in (0, 1)$ is a discount factor. The superscript $\pi$ in the expectation indicates that the actions $a_t$ are drawn from $\pi(h_t)$. The subscript $p$ indicates that the rewards and state transitions are induced by $p$.

In standard RL, we aim to find a policy that maximizes $V^\pi$. It is well-known that there exists a deterministic stationary policy $\pi : \mathcal{X} \to \mathcal{A}$ whose decisions depend only on the current state, that gives optimal $V^\pi$, and therefore one typically works in the space of stationary deterministic policies. Key to a dynamic-programming solution to the above problem is the use of a value function

$Q^\pi(x, a) := \mathbb{E}_p^\pi\left[\sum_{t=0}^\infty \gamma^t r_t \mid x_0 = x, a_0 = a\right]$, which satisfies the Bellman equation

$$\forall x, a, \ Q^\pi(x, a) = \sum_{r, x'} p(r, x'|x, a) \left[r + \gamma Q^\pi(x', \pi(x'))\right]. \tag{1}$$

The optimal value $Q^*(x, a) := Q^{\pi^*}(x, a)$ for any optimal policy $\pi^*$ satisfies the Bellman optimality equation

$$\forall x, a, \ Q^*(x, a) = \sum_{r, x'} p(r, x'|x, a) \left[r + \gamma \max_{a'} Q^*(x', a')\right]. \tag{2}$$

Furthermore, for any $Q$-function $Q \in \mathcal{Q} := \{q : \mathcal{X} \times \mathcal{A} \to \mathbb{R} \mid q(x, a) < \infty, \forall x, a\}$, one can show that the operator $\mathcal{T}^\pi$ defined by $\mathcal{T}^\pi Q(x, a) := \sum_{r, x'} p(r, x'|x, a)[r + \gamma Q(x', \pi(x'))]$ is a $\gamma$-contraction in the sup-norm $\|Q\|_\infty := \max_{x, a} |Q(x, a)|$ with fixed-point satisfying (1). One can therefore start with an arbitrary $Q$-function and repeatedly apply $\mathcal{T}^\pi$, or its stochastic approximation, to learn $Q^\pi$. An analogous operator $\mathcal{T}$ can also be shown to be a $\gamma$-contraction with fixed-point satisfying (2).

## 2.1 Static and Dynamic CVaR

The expected return $V^\pi$ is risk-neutral in the sense that it does not take into account the inherent variability of the return. In many application scenarios, one may prefer a policy that is more risk-averse, with better sensitivity to bad outcomes. In this work, we focus on the conditional value-at-risk (CVaR), which is a popular risk measure that satisfies the properties of being coherent (Artzner et al., 1999). The $\alpha$-level CVaR for a real-valued random variable $Z$, for $\alpha \in (0, 1]$, is given by (Rockafellar and Uryasev, 2000)

$$C_\alpha(Z) := \max_{s \in \mathbb{R}} \ s - \frac{1}{\alpha}\mathbb{E}[(s - Z)^+]$$

where $(x)^+ = \max\{x, 0\}$. Note that we are concerned with $Z$ that represents returns (the higher, the better), so this particular version of CVaR focuses on the lower tail of the distribution. In particular, the function $s \mapsto s - \frac{1}{\alpha}\mathbb{E}[(s - Z)^+]$ is concave in $s$ and the maximum is always attained at the $\alpha$-level quantile, defined as

$$q_\alpha(Z) := \inf\{s : \Pr(Z \le s) \ge \alpha\}.$$

For $\alpha = 1$, $C_\alpha$ reduces to the standard expectation. In the case $Z$ is absolutely continuous, we have the intuitive $C_\alpha(Z) = \mathbb{E}[Z|Z < q_\alpha(Z)]$.

Our target random variable is the total discounted return $Z^\pi := \sum_{t=0}^\infty \gamma^t r_t$ of a policy $\pi$, and our objective is to find a policy that maximizes $C_\alpha(Z^\pi)$, where the optimal CVaR is given by

$$\max_\pi \max_s \ s - \frac{1}{\alpha}\mathbb{E}_p^\pi[(s - Z^\pi)^+]. \tag{3}$$

In the context where $Z$ is accumulated over multiple time steps, the objective (3) corresponds to maximizing the so-called static CVaR. This objective is time-inconsistent in the sense that the optimal policy may be history-dependent and therefore non-Markov. This is, however, perfectly expected since the optimal behavior in the later time steps may depend on how much rewards have been accumulated thus far – more risky actions can be taken if one has already collected sufficiently large total rewards, and vice versa. From the point of view of dynamic programming, an alternative, time-consistent or Markovian version of CVaR may be more convenient. A class of such risk measures was proposed by Ruszczynski (2010), and we shall refer to this version of CVaR as the dynamic CVaR, defined recursively as[1]

$$\forall \pi, x, a, \quad D_{\alpha, 0}^\pi(x, a) := C_\alpha[r_t | x_t = x, a_t = a],$$
$$\forall \pi, x, a, T > 0, \quad D_{\alpha, T}^\pi(x, a) := C_\alpha[r_t + \gamma D_{\alpha, T-1}^\pi(x_{t+1}, \pi(x_{t+1})) | x_t = x, a_t = a],$$
$$\forall \pi, x, a, \quad D_\alpha^\pi(x, a) := \lim_{T \to \infty} D_{\alpha, T}^\pi(x, a).$$

---

[1]We use a slightly different definition from that in (Ruszczynski, 2010), but conceptually they are essentially the same.

It can be shown (Ruszczynski, 2010) that there exists a stationary deterministic optimal policy $\pi^*$, maximizing $D_\alpha^\pi(x, a)$ for all $x, a$, whose dynamic CVaR is given by $D_\alpha^* := D_\alpha^{\pi^*}$. In particular, the operator $\mathcal{T}_\alpha^D$ defined by

$$\mathcal{T}_\alpha^D D(x, a) := C_\alpha[r_t + \gamma \max_{a'} D(x_{t+1}, a') | x_t = x, a_t = a] \tag{4}$$

for $D \in \mathcal{Q}$ is a $\gamma$-contraction in sup-norm with fixed-point satisfying

$$\forall x, a, \quad D_\alpha^*(x, a) = C_\alpha[r_t + \gamma \max_{a'} D_\alpha^*(x_{t+1}, a') | x_t = x, a_t = a]. \tag{5}$$

Despite its theoretical properties, the dynamic CVaR is hard to interpret. Moreover, from a practical point of view, it can be overly optimistic in certain cases and overly conservative in other cases. We illustrate with some examples in Section 2.2. In such cases it may be favorable to use the static CVaR. Bäuerle and Ott (2011) suggest an iterative process that, in theory, can be used to solve for the optimal static CVaR policy. The approach is based on (3):

1. For a fixed $\hat{s}$, one can solve for the optimal policy with respect to $\max_\pi \mathbb{E}[-(\hat{s} - Z^\pi)^+]$.
2. For a fixed $\pi$, the optimal $s$ is given by the $\alpha$-level quantile of $Z^\pi$.
3. Repeat until convergence.

Step one above can be done by solving an augmented MDP $\widetilde{\mathcal{M}}$ with states $\tilde{x} = (x, s) \in \mathcal{X} \times \mathbb{R}$, where $s$ is a moving threshold keeping track of the accumulated rewards so far. [2] In particular, this MDP has no rewards (except in terminal states) and state transitions are given by $\tilde{p}(0, (x', \frac{s-r}{\gamma}) | (x, s), a) := p(r, x' | x, a)$. Solving $\widetilde{\mathcal{M}}$ directly using model-free RL, however, can result in poor sample efficiency since each example $(x, a, r, x')$ may need to be experienced many times under different threshold $s$. Furthermore, there is a question of updating $\hat{s}$. In this work, we propose an alternative solution using the approach of distributional RL.

## 2.2 Distributional RL

In standard RL, one typically learns the $Q^\pi(x, a)$ value for each $(x, a)$ through some form of temporal-difference learning (Sutton and Barto, 2018). In distributional RL (Bellemare et al., 2017), one instead tries to learn the entire distribution of possible future return $Z^\pi(x, a)$ for each $(x, a)$. The $Q$-value can then be extracted by simply taking the expectation $Q^\pi(x, a) = \mathbb{E}[Z^\pi(x, a)]$.

The objects of learning are distribution functions $U \in \mathcal{Z} := \{Z : \mathcal{X} \times \mathcal{A} \to \mathcal{P}(\mathbb{R}) \mid \mathbb{E}[|Z(x, a)|^q] < \infty, \forall x, a, q \geq 1\}$. For any state-action pair $(x, a)$, we use $U(x, a)$ to denote a random variable with the respective distribution. Let $\widetilde{\mathcal{T}}^\pi$ be the distributional Bellman operator on $\mathcal{Z}$ such that

$$\widetilde{\mathcal{T}}^\pi U(x, a) \overset{D}{:=} R + \gamma U(X', \pi(X'))$$

where $\overset{D}{=}$ denotes equality in distribution, generated by the random variables $R, X'$ induced by $p(r, x' | x, a)$. We use the notation $\widetilde{\mathcal{T}}$ instead of $\mathcal{T}$ when referring to a distributional operator, where $\widetilde{\mathcal{T}}^\pi U(x, a)$ is a random variable. Bellemare et al. (2017) show that $\widetilde{\mathcal{T}}^\pi$ is a $\gamma$-contraction in $\mathcal{Z}$ in the following distance metric

$$d(U, V) := \sup_{x, a} \omega(U(x, a), V(x, a))$$

where $\omega$ is the 1-Wasserstein distance between the distributions of $U(x, a)$ and $V(x, a)$. Furthermore, the operator $\widetilde{\mathcal{T}}$ defined by

$$\widetilde{\mathcal{T}} U(x, a) \overset{D}{:=} R + \gamma U(X', A'), \quad A' = \arg\max_{a'} \mathbb{E}[U(X', a')] \tag{6}$$

can be shown to be a $\gamma$-contraction in $\mathcal{Q}$ (not necessarily in $\mathcal{Z}$) in sup-norm under element-wise expectation, i.e.,

$$\|\mathbb{E}\widetilde{\mathcal{T}}U - \mathbb{E}\widetilde{\mathcal{T}}V\|_\infty \leq \gamma \|\mathbb{E}U - \mathbb{E}V\|_\infty,$$

---

[2]The results by Bäuerle and Ott (2011) apply generally to any Borel state-action space, with mild technical conditions. Our finite state-action setting is a special case where such conditions are trivially statisfied. Also, since we assume a countable and bounded reward set, the resulting augmented MDP $\widetilde{\mathcal{M}}$ has countable state space, which simplifies the technical presentation of our main ideas.

where $\mathbb{E}\widetilde{\mathcal{T}}U \in \mathcal{Q}$ such that $\mathbb{E}\widetilde{\mathcal{T}}U(x,a) := \mathbb{E}[\widetilde{\mathcal{T}}U(x,a)]$, and $\mathbb{E}U$, $\mathbb{E}V$, $\mathbb{E}\widetilde{\mathcal{T}}V$ all similarly defined. In general, $\widetilde{\mathcal{T}}$ is not expected to be a contraction in the space of distributions $\mathcal{Z}$ for the obvious reason that multiple optimal policies can have very different distributions of the total return even though they all have the same expected total return.

Since one keeps the full distribution instead of just the expectation, a natural way to exploit this is to extract more than just the expectation from each distribution. In particular, in (6), one can select the action $a'$ based on $C_\alpha[U(x',a')]$ instead of $\mathbb{E}[U(x',a')]$ for a risk-averse strategy. This strategy is proposed by Dabney et al. (2018a) and Keramati et al. (2020), among others, which we now refer to as the Markov action-selection strategy:

$$\widetilde{\mathcal{T}}_\alpha^D U(x,a) :\overset{D}{=} R + \gamma U(X', A'), \qquad A' = \arg\max_{a'} C_\alpha[U(X', a')]. \tag{7}$$

One may guess that this converges to the optimal dynamic CVaR policy satisfying (5). We now show that in general this is not true.

**Proposition 1.** *In general, the distributional Bellman operator with the Markov action-selection strategy, $\widetilde{\mathcal{T}}_\alpha^D$, converges to neither the optimal dynamic-CVaR nor the optimal static-CVaR policies, even if the optimal CVaR policy is stationary and Markov.*

*Proof.* We prove by counterexamples. We use the notation $\{(p_1; r_1), (p_2; r_2), \ldots\}$ to denote a random variable that takes value $r_1$ with probability $p_1$, $r_2$ with probability $p_2$ and so on. For simplicity we assume $\gamma = 1$, but the examples work for any $\gamma$ by simple scaling of the rewards. Let $p, \epsilon$ such that $0 < \epsilon \ll p < 1 - \epsilon$. Assume that the chosen CVaR level $\alpha$ is such that $p^2 + \epsilon < \alpha < p$. We only consider deterministic policies since both dynamic and static CVaR admit deterministic optimal policies.

For the MDP in Fig.1 (a), let $X_1$ be the initial state. There are only two possible policies: choosing either action $A_1$ or $A_2$ in state $X_1$. It is easy to see that for $\alpha < p$, $D_\alpha^*(X_2) = 0$ and $D_\alpha^*(X_1) = 0$, and therefore the optimal dynamic-CVaR policy is $A_2$. However, $\widetilde{\mathcal{T}}_\alpha^D U$ for any $U \in \mathcal{Z}$ converges to $U^*(X_1, A_1) = \{(p^2; 0), (1 - p^2; 1)\}$ so $C_\alpha[U^*(X_1, A_1)] > \epsilon$ for $\alpha > p^2 + \epsilon$ and therefore $A_1$ will be selected policy.

For the MDP in Fig.1 (c), again let $X_1$ be the initial state. The only possible policies correspond to choosing either $A_1$ or $A_2$ in $X_2$. It is easy to see that $\widetilde{\mathcal{T}}_\alpha^D U$ for any $U \in \mathcal{Z}$ converges to $U^*(X_2, A_1) = \{(p; 0), (1 - p; 1)\}$ and $U^*(X_2, A_2) = \{(1; \epsilon)\}$ respectively and therefore $A_2$ will be the chosen action for $X_2$ whenever $\alpha < p$. By (7), this in turn means that $U^*(X_1) = \{(p; \epsilon), (1 - p; 1)\}$, thus $C_\alpha[(U^*(X_1)] = \epsilon$. The optimal static-CVaR policy, however, is to always choose $A_1$ in $X_2$ (i.e. stationary and Markov), and will result in $C_\alpha^* > \epsilon$ for $\alpha > p^2 + \epsilon$. $\qquad\square$

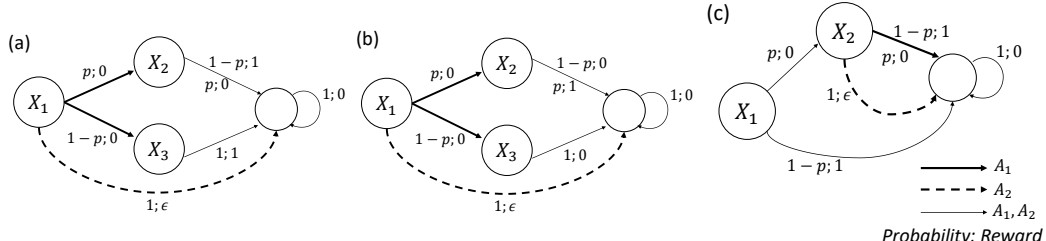

Figure 1: MDPs with actions $A_1, A_2$ and initial state $X_1$, and consider $p^2 < \alpha < p$. (a) Example where $D_\alpha^*$ underestimates the true CVaR. (b) Example where $D_\alpha^*$ overestimates the true CVaR. (c) Example where the Markov action-selection strategy results in underestimating the true CVaR.

## 2.3 Distributional RL for CVaR

It is now natural to ask whether we can properly optimize for the static CVaR while still staying within the framework of distributional RL. Recall that it is possible to optimize for the static CVaR by solving an augmented MDP $\widetilde{\mathcal{M}}$ as part of an iterative process. We recall some results regarding $\widetilde{\mathcal{M}}$. Let $\tilde{\pi}$ be a policy in $\widetilde{\mathcal{M}}$ and $W^{\tilde{\pi}}(x, s, a)$ be the $Q$-value of $\tilde{\pi}$ in $\widetilde{\mathcal{M}}$. Let $Z^{\tilde{\pi}}(x, s, a)$ be the

random variable representing the total return $\sum_{t=0}^{\infty} \gamma^t r_t$ in the original MDP $\mathcal{M}$ when following $\tilde{\pi}$ after executing $a$ at $x$ (assuming the threshold $s$). Let $Z^{\tilde{\pi}}(x, s) = Z^{\tilde{\pi}}(x, s, \tilde{\pi}(x, s))$. Note that $\tilde{\pi}$ is a non-Markov policy in $\mathcal{M}$ and executing it in $\mathcal{M}$ requires keeping track of the reward history with $s_{t+1} = \frac{s_t - r_t}{\gamma}$. From (Bäuerle and Ott, 2011), we have that:

$$W^{\tilde{\pi}}(x, s, a) = \mathbb{E}_{z \sim Z^{\tilde{\pi}}(x,s,a)}\left[-(s-z)^+\right]. \tag{8}$$

The key observation here is that we can extract the $Q$-value of $\widetilde{\mathcal{M}}$ from the distribution $Z^{\tilde{\pi}}$ through (8). The problem, however, is that in general the distribution $Z^{\tilde{\pi}}(x, s, a)$ for a fixed $(x, a)$ may vary depending on $s$. This means that keeping only a single value distribution for each $(x, a)$ precludes the extraction of the optimal CVaR policy in general. However, for the special case where there exists a stationary Markov optimal CVaR policy, we now show that this is possible.

Let $\tilde{\pi}^*$ be an optimal $\alpha$-CVaR policy that is stationary and Markov in $\mathcal{M}$. By definition of being stationary and Markov, there exists $\pi^*$ such that $\tilde{\pi}^*(x, s) = \pi^*(x)$ for every $(x, s)$ pair encountered when running $\tilde{\pi}^*$ starting from $(x_0, s^*)$ with $s^* = q_\alpha(Z^{\tilde{\pi}^*}(x_0, s^*))$. Let $\mathcal{S}^*(x)$ be the set $\{s : \exists t \geq 0, \Pr^{\tilde{\pi}^*}[\tilde{x}_t = (x, s) | \tilde{x}_0 = (x_0, s^*)] > 0\}$. In words, $\mathcal{S}^*(x)$ is the set of all possible thresholds $s$ that one could encounter at state $x$, when following $\tilde{\pi}^*$ from $x_0$. Since we assume that the set of rewards $\mathcal{R}$ is countable, the set of possible $s$ is also countable so the probability in the definition of $\mathcal{S}^*(x)$ is well-defined. It follows that $Z^{\tilde{\pi}^*}(x, s) \overset{D}{=} Z^{\pi^*}(x, \pi^*(x))$ for all $(x, s)$ where $s \in \mathcal{S}^*(x)$. We call $\mathcal{S}^*(x)$ the *active set* of $x$ under $\tilde{\pi}^*$.

For any distribution function $U \in \mathcal{Z}$, let

$$W^U(x, s, a) := \mathbb{E}_{z \sim U(x,a)}[-(s-z)^+]. \tag{9}$$

Suppose that we are given $U \in \mathcal{Z}$ such that $U(x, a) \overset{D}{=} Z^{\pi^*}(x, a)$ for all $(x, a) \in \mathcal{X} \times \mathcal{A}$. We then execute a policy based on $U$ using Algorithm 1.

---

**Algorithm 1** Policy execution for static CVaR for one episode

---

Input: $\gamma \in (0, 1)$, $\alpha \in (0, 1]$, $U \in \mathcal{Z}$

    1. $x \leftarrow x_0$

    2. $a \leftarrow \arg\max_{a'} C_\alpha[U(x, a')]$

    3. $s \leftarrow q_\alpha(U(x, a))$

    4. While $x$ not terminal state,

        (a) Execute $a$ in $x$, observe reward $r$ and next state $x'$

        (b) $x \leftarrow x'$

        (c) $s \leftarrow \frac{s-r}{\gamma}$

        (d) $a \leftarrow \arg\max_{a'} W^U(x, s, a')$

---

**Proposition 2.** *Let $\pi^*$ be a (stationary and Markov) $\alpha$-CVaR optimal policy in $\mathcal{M}$. Assume that $\pi^*$ is unique in $\mathcal{M}$. Running Algorithm 1 with $U \in \mathcal{Z}$ such that $U(x, a) \overset{D}{=} Z^{\pi^*}(x, a)$ for all $(x, a) \in \mathcal{X} \times \mathcal{A}$ results in executing $\pi^*$.*

*Proof.* We show by induction that for every state $x_t$, Algorithm 1 executes $\pi^*(x_t)$. At $x_0$, action is selected by the $\alpha$-level CVaR of $U(x, a') \overset{D}{=} Z^{\pi^*}(x, a')$. By the optimality and uniqueness of $\pi^*$, $\pi^*(x_0)$ will be selected.

Suppose that Algorithm 1 executes $\pi^*(x_t)$ for $t = 0, 1, \ldots, (T-1)$. In $\widetilde{\mathcal{M}}$, the state $(x_T, s_T)$ is such that $s_T \in \mathcal{S}^*(x_T)$ so $\tilde{\pi}^*(x_T, s_T) = \pi^*(x_T)$ and $Z^{\tilde{\pi}^*}(x_T, s_T, \tilde{\pi}^*(x_T, s_T)) \overset{D}{=} Z^{\pi^*}(x_T, \pi^*(x_T))$. We have that for each $a \in \mathcal{A}$,

$$W^U(x_T, s_T, a) = \mathbb{E}_{z \sim U(x_T,a)}[-(s_T - z)^+] \overset{(a)}{\leq} \mathbb{E}_{z \sim Z^{\tilde{\pi}^*}(x_T, s_T, a)}[-(s_T - z)^+]$$

$$= W^{\tilde{\pi}^*}(x_T, s_T, a) \overset{(b)}{\leq} W^{\tilde{\pi}^*}(x_T, s_T, \tilde{\pi}^*(x_T, s_T)) \overset{(c)}{=} W^U(x_T, s_T, \pi^*(x_T))$$

where the inequality (a) is due to $\tilde{\pi}^*$ being the optimal policy in $\widetilde{\mathcal{M}}$, and (c) is due to the observation that $s_T \in \mathcal{S}^*(x_T)$ above. By the uniqueness of $\pi^*$, only $\pi^*(x_T)$ can achieve equality at both (a) and (b) and therefore it will be the selected action at $T$. $\qquad\square$

Proposition 2 establishes the fact that one can execute a CVaR-optimal policy from a distribution function $U \in \mathcal{Z}$ by tracking the states in the augmented MDP $\widetilde{\mathcal{M}}$. Our objective is to learn such a distribution function using distributional RL. Figure 1(c) and Proposition 1 show that even for an MDP with stationary Markov optimal-CVaR policy, $\widetilde{\mathcal{T}}_\alpha^D$ as defined in (7) will not converge to the optimal CVaR policy. We now propose an alternative distributional Bellman operator for CVaR. Given a mapping $\psi : \mathcal{X} \to \mathbb{R}$, define

$$\widetilde{\mathcal{T}}_\psi U(x, a) :\stackrel{D}{=} R + \gamma U(X', A'), \quad A' = \arg\max_{a'} W^U\left(X', \frac{\psi(x) - R}{\gamma}, a'\right) \qquad (10)$$

where $W^U(x, s, a)$ is as defined in (9). Note that $A'$ is a function of random variables $X'$ and $R$, induced by $p(r, x'|x, a)$. We have the following result on $\widetilde{\mathcal{T}}_\psi$:

**Proposition 3.** *Let $\tilde{\pi}^*$ be a unique, optimal $\alpha$-CVaR policy that is stationary and Markov in $\mathcal{M}$. Choose $\psi : \mathcal{X} \to \mathbb{R}$ such that $\psi(x) \in \mathcal{S}^*(x)$ for all $x$, where $\mathcal{S}^*(x)$ is the active set of $x$ under $\tilde{\pi}^*$. Let $\pi^*$ be the policy in $\mathcal{M}$ such that $\tilde{\pi}^*(x, s) = \pi^*(x)$ for all $(x, s)$ where $s \in \mathcal{S}^*(x)$. Then $\widetilde{\mathcal{T}}_\psi Z^{\pi^*}(x, \pi^*(x)) \stackrel{D}{=} Z^{\pi^*}(x, \pi^*(x))$.*

*Proof.* The action-selection strategy in Algorithm 1 is exactly that of $\widetilde{\mathcal{T}}_\psi$. Proposition 2 shows that applying $\widetilde{\mathcal{T}}_\psi$ on $Z^{\pi^*}(x, \pi^*(x))$ will always select the same mixture of distributions since $\frac{\psi(x) - r}{\gamma} \in \mathcal{S}^*(x')$ for all $x, r, x'$ with $p(r, x'|x, \pi^*(x)) > 0$. $\qquad\square$

Proposition 3 says nothing about $\widetilde{\mathcal{T}}_\psi Z^{\pi^*}(x, a)$ if $a \neq \pi^*(x)$. We need a stronger condition for this.

**Proposition 4.** *Let $\tilde{\pi}^*$ and $\psi$ be as defined in Proposition 3. Let $\mathcal{S}^{**}(x) = \{\frac{\psi(x') - r}{\gamma} : \exists x', a, \ p(r, x|x', a) > 0\}$. If there exists $\pi^*$ in $\mathcal{M}$ such that $\tilde{\pi}^*(x, s) = \pi^*(x)$ for all $(x, s)$ where $s \in \mathcal{S}^{**}(x)$, then $Z^{\pi^*}$ is a fixed-point of $\widetilde{\mathcal{T}}_\psi$.*

*Proof.* With the same reasoning as in the proof of Proposition 2 and 3, it can be seen that $\widetilde{\mathcal{T}}_\psi = \widetilde{\mathcal{T}}^{\pi^*}$. It follows that $\widetilde{\mathcal{T}}_\psi Z^{\pi^*} = \widetilde{\mathcal{T}}^{\pi^*} Z^{\pi^*} = Z^{\pi^*}$. $\qquad\square$

Even if $Z^{\pi^*}$ is a fixed-point of $\widetilde{\mathcal{T}}_\psi$, it is an open question whether $\widetilde{\mathcal{T}}_\psi U$ will converge to $Z^{\pi^*}$ for all $U \in \mathcal{Z}$. On simple MDPs with known optimal policies, we have observed that $\widetilde{\mathcal{T}}_\psi$ not only converges, but can extract both stationary and even non-stationary CVaR-optimal policies. Some of these examples are included in the supplementary material.

## 3  Algorithm

Our proposed algorithm is based on distributional Q-learning using quantile regression (Dabney et al., 2018b). It can be readily adapted to other variants of distributional RL. In this approach, each distribution $U(x, a)$ is approximated by $N$ quantiles $\theta_i(x, a)$, $i = 1 \ldots N$, each corresponds to a quantile level $\hat{\tau}_i = \frac{i - 0.5}{N}$. The quantile function $q_\alpha(U(x, a))$ for any $(x, a)$ can therefore be conveniently extracted from $\theta(x, a)$. Similarly, $W^U(x, s, a)$ can be computed by $\frac{1}{N} \sum_{i=1}^{N} [-(s - \theta_i(x, a))^+]$. Given $\theta$, one can then execute Algorithm 1.

For training, one typically represents $\theta$ by a neural-network that maps input $x$ to output $\theta_i(x, a)$ for each $i$ and $a$. The neural-network is updated based on gradients computed using sampled mini-batches from a buffer that stores actual transition examples during training runs. In our proposed approach, each stored example includes $x_k$, $s_k$, $a_k$, $r_k$ and $x_k'$. The gradients are computed with respect to a quantile-regression loss function $\rho_\tau(u) = u(\tau - \delta_{u<0})$ where $\delta_{u<0} = 1$ if $u < 0$ and 0 otherwise. Algorithm 2 shows the main algorithm for computing the loss over a mini-batch containing $m$ transition samples. Notice that Algorithm 2 employs the standard practice of using a target network

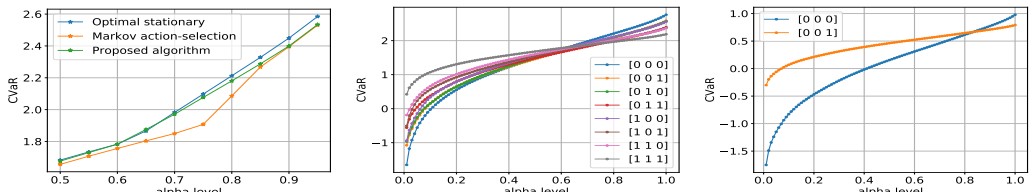

Figure 2: Left: Comparison with optimal stationary policies. Middle: Ground truth CVaR at $x_0$. Right: Ground truth CVaR at $x_2$.

$\theta'$ which is periodically updated from $\theta$. The key difference from the ordinary quantile-regression distributional Q-learning is our target action selection strategy for choosing $a'_k$ (Step 1(a)). For other implementation details, we refer the reader to (Dabney et al., 2018b).

---

**Algorithm 2** Quantile Regression Distributional Q-Learning for static CVaR

Input: $\gamma$, $\theta$, $\theta'$, mini-batch $(x_k, s_k, a_k, r_k, x'_k)$ for $k = 1 \dots m$

    1. For each $k = 1 \dots m$,

        (a) $a'_k \leftarrow \arg\max_{a'} W^{\theta'}(x'_k, \frac{s_k - r_k}{\gamma}, a')$

        (b) $\widetilde{\mathcal{T}}_\psi \theta_j(x_k, a_k) \leftarrow r_k + \gamma \theta'_j(x'_k, a'_k) \quad , j = 1 \dots N$

    2. $\mathcal{L} \leftarrow \frac{1}{m} \sum_{k=1}^m \frac{1}{N^2} \sum_{i,j} \rho_{\hat{\tau}_i}(\widetilde{\mathcal{T}}_\psi \theta_j(x_k, a_k) - \theta_i(x_k, a_k))$

    3. Output $\nabla\mathcal{L}$.

---

## 4 Empirical Results

Unless otherwise stated, we implement Algorithm 1 and 2 and represent our policies using a neural network with two hidden layers, with ReLU activation. All our experiments use Adam as the stochastic gradient optimizer. For each action, the output consists of $N = 100$ quantile values. Additional details and results, as well as the complete code to reproduce our results can be found in the supplementary material. [3]

### 4.1 Synthetic Data

We first evaluate our proposed algorithm in a simple task where we know the optimal stationary policy for any CVaR level. The MDP has 4 states $x_0, x_1, x_2, x_3$ where state $x_0$ is the initial state and $x_3$ is a terminal state. Each state has two actions $a_0$ and $a_1$. Action $a_0$ generates an immediate reward following a Gaussian $\mathcal{N}(1, 1)$ and action $a_1$ has immediate reward $\mathcal{N}(0.8, 0.4^2)$. Clearly, $a_0$ gives a better expected reward but with higher variance. Each action always moves the state from $x_i$ to $x_{i+1}$. We use $\gamma = 0.9$ for this task. For $\alpha > 0.63$, the optimal stationary policy is to choose action $a_0$ in all states, while for $\alpha < 0.62$, the optimal stationary policy is to choose action $a_1$ in all states. We compare our proposed algorithm with the optimal stationary policy at various levels of CVaR. Figure 2 (left) shows the results.

The Markov action-selection strategy corresponds to $\widetilde{\mathcal{T}}_\alpha^D$, which is the strategy proposed by Dabney et al. (2018a). Our proposed strategy corresponds to Algorithm 1 and 2. We clearly see that the proposed strategy outperforms the Markov strategy at all tested CVaR levels. Further insights are revealed in Figure 2 (middle and right). These are the ground truth CVaR values for all the stationary policies, where $[1\,0\,0]$ means always choosing action $a_1$ in $x_0$ and $a_0$ in the next two states. Notice the switching point around $\alpha = 0.625$ in the middle plot and around $\alpha = 0.83$ in the right plot. The Markov action-selection strategy will choose action $a_1$ in $x_2$ for $\alpha < 0.83$ since this is the better

---

[3]We omit any comparison with policy-gradient methods for CVaR, whose gradient estimation involves the use of the Lagrangian. A discussion of the pros and cons of value-based vs policy-gradient methods would distract too much from our main focus here.

action if one ignores the rewards collected since the beginning. However, this results in a rather conservative strategy since the optimal strategy should still favor $a_0$ in $x_2$ for $\alpha > 0.625$.

## 4.2 Option Trading

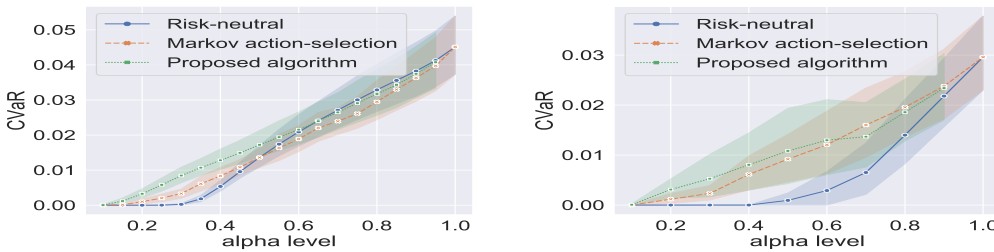

Figure 3: Test results on synthetic (Left) and real (Right) prices.

We evaluate our proposed algorithm on the non-trivial real-world task of option trading, commonly used as a test domain for risk-sensitive RL (Li et al., 2009; Chow and Ghavamzadeh, 2014; Tamar et al., 2017). In particular, we tackle the task of learning an exercise policy for American options. This can be formulated as a discounted finite-horizon MDP with continuous states and two actions. The state $x_t$ includes the price of a stock at time $t$, as well as the number of steps to the maturity date, which we set to $T = 100$. The first action, "hold", will always move the state one time step forward with zero reward, while the second action, "execute", will generate an immediate reward $\max\{0, K - x_t\}$ and enter a terminal state. $K$ is the strike price. In our experiments, we use $K = 1$ and always normalize the prices such that $x_0 = 1$. At $t = T - 1$, all actions will be interpreted as "execute". We set $\gamma = 0.999$, which corresponds to a non-zero daily risk-free interest rate.

We use actual daily closing prices for the top 10 Dow components from 2005 to 2019. Prices from 2005-2015 are used for training and prices from 2016-2019 for testing. To allow training on unlimited data, we follow (Li et al., 2009) and create a stock price simulator using the geometric Brownian motion (GBM) model. The GBM model assumes that the log-ratio of prices follows a Gaussian distribution $\log \frac{x_{t+1}}{x_t} \sim \mathcal{N}(\mu - \sigma^2/2, \sigma^2)$ with parameters $\mu$ and $\sigma$, which we estimate from the real training data.

For each algorithm, each stock and each CVaR level, we trained 3 policies using different random seeds. The policies are then tested on the synthetic data (generated using the same training model) for 1000 episodes. The policies are further tested on the real data, using 10 episodes, each with 100 consecutive days of closing prices, covering the 4 years of test period. All results are averaged over the 3 policies and over the 10 stocks.

Figure 3 shows the test results on synthetic (Left) as well as real data (Right). The proposed strategy clearly performs better across various CVaR levels. The gap is significant at lower $\alpha$ levels. Also included are the results from the risk-neutral strategy, trained using $\alpha = 1$, and tested on all $\alpha$ values. This corresponds to the standard action-selection strategy based on the expected return, which performs badly at low $\alpha$ levels.

## 4.3 Atari Games

Atari games have been a popular domain for benchmarking deep RL algorithms. Dabney et al. (2018a) evaluated risk-sensitive approaches on several games but acknowledged that qualitatively the outcome did not always match their expectations. We believe that the difficulty in interpreting the results is due to the many confounding factors at play, e.g. total training time, exploration strategies, learning rate etc. Here, we compare the effect of our proposed approach with that of the Markov action-selection strategy in terms of the total discounted return starting from the initial state. This means that for discount factor $\gamma < 1$, the initial stage of the game plays a much more significant role than later stages.

We use the same implicit quantile networks architecture as in (Dabney et al., 2018a). We use the implementation by Fujita et al. (2021) and made the slight modifications needed for Algorithms 1

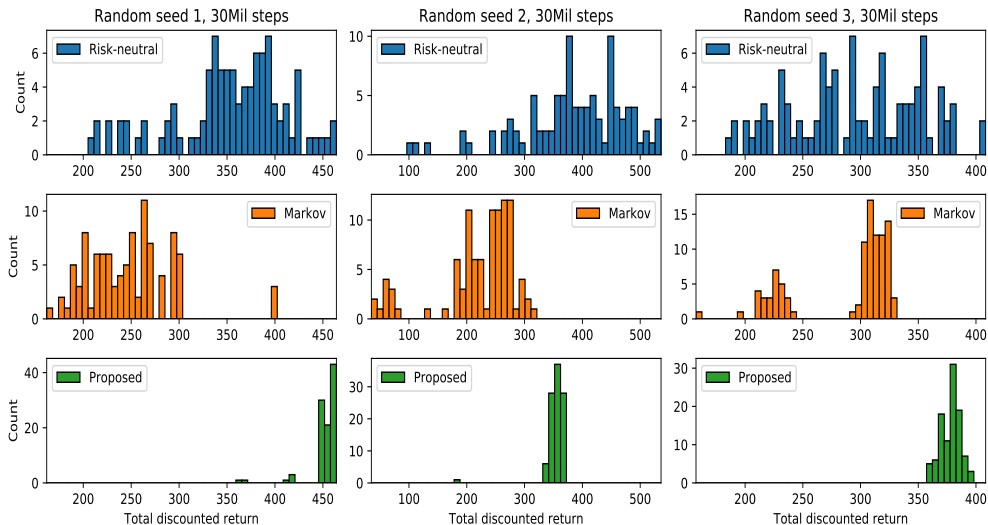

Figure 4: Results on Atari game Asterix after 30-million training steps. Each plot is a histogram of the discounted returns over 100 evaluation episodes.

and 2. We optimize for 0.25-CVaR for both risk-averse strategies, with $\gamma = 0.99$. Figure 4 shows the performance of the learned policies after going through 30 million steps of the game Asterix, across 3 random seeds. Each plot is a histogram of the returns over 100 evaluation runs. We can clearly see that the proposed approach achieves a more concentrated distribution of returns compared to both the risk-neutral and the Markov strategies, and overall with better 0.25-CVaR. In this particular example, we observe that the Markov strategy actually performs worse than the risk-neutral strategy, both in terms of the mean and the 0.25-CVaR, although the resulting policies still have more concentrated distribution of returns in the test runs. The outcomes may change if we train for a larger number of steps, but we leave it to future work for more extensive empirical investigations. Additional results can be found in the supplementary material.

## 5 Conclusion and Future Work

This work points out a problem that arises in existing methods that extend distributional RL to the CVaR risk measure, and proposes an approach with better theoretical properties. We have shown that the proposed approach for learning a CVaR-optimized policy works in a variety of task domains and can produce better risk-averse policies. Furthermore, the proposed algorithms can be easily incorporated into existing distributional RL frameworks.

On simple tasks for which we know the optimal CVaR policies, we have shown empirically that our approach managed to converge to these optimal policies. Theoretically, however, the convergence properties of the proposed $\widetilde{\mathcal{T}}_\psi$ operator remain unknown in the general settings and we believe this is an interesting problem for future works. In terms of the practical use of the proposed algorithms, one open problem is regarding the repeated use of the stored threshold $s$ in the replay buffer after a significant change in the return distribution of the policy. We believe that a more adaptive way of utilizing $s$ during training can lead to improvement in the learning efficiency in terms of speed and stability. In terms of exploration, an optimistic bias such as that proposed by Keramati et al. (2020) may also be beneficial to our approach.

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
