# OpenReview forum: "Distributional Reinforcement Learning for Risk-Sensitive Policies"
_NeurIPS.cc/2022/Conference — NeurIPS 2022 Accept_

### Official Review · Reviewer_XVo8 · 2022-07-06

**Rating:** 7
**Confidence:** 4
**Soundness:** 3 good
**Presentation:** 4 excellent
**Contribution:** 4 excellent

**Summary:**

The authors tackle the challenge of learning a risk averse policy (CVaR) in a distributional RL setting.
While prior works have presented a naive method for optimizing the CVaR, this is shown (both theoretically and practically) to not optimize the correct objective.
In this work, they present a theoretically sound method for iteratively learning a CVaR maximizing policy and back these claims with experiments.

**Questions:**

1) The comparison to Dabney et. al. 2018 (a) seems slightly misleading.
The goal of a risk-sensitive method is to maximize the lower-tail of the returns. Dabney et. al. 2018 (a) might not optimize the CVaR properly, but their results do show competitive performance in ATARI domains (often outperforming mean optimization).

In these experiments, we see that Markov selection is under-performing compared to risk-neutral.

What happens in simpler domains / when the algorithm is run using the same parameters as Dabney et. al. 2018 (a) and for a longer period of time (commonly 200m steps)?

2) What happens in other domains? does this method still work or is it limited to Asterix?

**Limitations:**

-

**Strengths And Weaknesses:**

Strengths:

- Present a theoretically backed 1-step iterative update scheme for learning a CVaR maximizing policy in a distributional RL setting.
- Present a comparison and analysis of prior methods showing that naive application of the 1-step CVaR does not solve the objective.
- Experiments on synthetic data + option trading + ATARI.

Weakness:

- While this work performs analysis on atari, this analysis is very limited. While I appreciate the simpler experiments that enable a more theoretical analysis of the method, complex domains enable a more convincing empirical justification.
I'd like to see more experiments in such domains. If compute is an issue, the authors can always resort to tasks such as continuous control / minatari / cartpole / and other simpler tasks.
- Paper lacks a conclusion and future work section. While there are no rules that require such structure, I do feel that this is warranted and well accepted in the community. I'd propose to move some proofs to the appendix in order to allocate room for discussion.

---

> ### Author Response · Authors · 2022-07-30
> **Re: Official Review of Paper5987 by Reviewer XVo8**
>
> Thanks for the helpful review!
>
> Question 1 & 2:
>
> We did have empirical results on other domains in the Appendix: Modified Puddle World, Noisy Lunar Lander and an additional Atari game Qbert.
>
> On simpler domains (e.g. Modified Puddle World) we indeed can observe the anticipated outcomes: both Markov and the proposed strategies achieve better lower tails in the return distribution (while performing worse in the overall expected return). The proposed strategy tends to achieve the best performance in terms of the target objective (the targeted alpha-level CVaR).
>
> On the Atari domains, the conclusion is more mixed (similarly observed by Dabney et al). We do see instances where the outcomes are as expected (like in simpler domains above) but not always. We agree that running the training for 200 million steps (instead of 30 million) may result in different outcomes. While our limited computational resources prevent us from fully exploring the empirical aspects, we hope that the included code (for full replication of our results) will allow other researchers to probe this further.
>
> Conclusion/future work
>
> We agree that a discussion/conclusion/future work section can be helpful.  We will move some contents to the Appendix and add this section in the final version.

---

### Official Review · Reviewer_wokp · 2022-07-09

**Rating:** 6
**Confidence:** 3
**Soundness:** 3 good
**Presentation:** 2 fair
**Contribution:** 3 good

**Summary:**

The authors study the problem of learning risk-sensitive policies in the distributional RL setting. They adopt Conditional Value at Risk (CVaR) as the risk measure throughout the paper. They begin with a problem overview and discuss the differences between static and dynamic CVaR problem formulations. After introducing the requisite background on distributional RL, the authors propose a naïve method of learning a risk-sensitive policy, called the Markov action-selection strategy, where the action selection procedure aims to maximize the $\alpha$-CVaR over return distributions at a given state. It is then shown that a distributional Bellman operator employing the Markov action-selction strategy does not converge to a dynamic or static CVaR optimal policy, even if the optimal CVaR policy is stationary and Markov.

With this result, the authors’ aim is thus to determine if a static CVaR policy may be properly optimized in the distributional RL framework. They begin by constructing an augmented MDP from the original, with the addition of a thresholding parameter. This parameter is initially set equal to the $\alpha$-quantile of the initial return distribution and is repeatedly updating by subtracting off observed rewards, thereby effectively keeping a history of the “return-to-go” to reach the initial threshold. The introduction of this new parameter poses a new problem: it is no longer sufficient to retain a single value distribution for the original MDPs state-action pairs, since the value distributions in the augmented MDP depend on the threshold parameter, which may be observed to be many different values for any given state-action pair. To navigate this problem, the authors assume the existence of a CVaR optimal augmented policy that is stationary and Markov in the original MDP, and show that the stationarity and Markov properties allow for one to recover an optimal CVaR policy in the original MDP. They go on to show that given the optimal original policy’s value distribution, Algorithm 1 executes the optimal original policy by keeping track of the thresholding parameter’s history. It is shown thereafter that the action selection procedure in Algorithm 1 may be used in a distributional Bellman operator, and that this new operator has a fixed point that is equal to the return distribution associated with the optimal CVaR policy from the original un-augmented MDP. The authors note here that it is still an open question whether the new Bellman operator is a contractive operator on the entire space of return distributions i.e. that starting from any return distribution, the repeated application of the operator would result in the eventual fixed point that is equal to the return distribution of the CVaR optimal policy.

The authors then present an algorithm that optimizes a neural network policy using the newly proposed distributional Bellman operator. Experiments are presented showing that the proposed algorithm outperforms the Markov action-selection and risk neutral policies in a synthetic data experiment, an options trading experiment, and in Atari games.


**Questions:**

N/A

**Limitations:**

-The paper does discuss the theoretical limitations of the work, namely the absence of a proof of contractivity of the proposed Bellman operator.

**Strengths And Weaknesses:**

Strengths:

-The paper introduces novel and interesting theoretical results that advance the theory of CVaR distributional RL

-The research presented in the paper lays groundwork for important future research, e.g. proving the contractive property of the proposed Bellman operator.

-The paper does a very good job of clearly stating the need, contribution, and relevance of the work. This is reinforced by the detailed background provided in Section 2


Weaknesses:

-The paper ends abruptly and has no concluding remarks.

-The discussion and analysis of the empirical results is limited. There is very little explanation offered that would allow the reader to construct an intuitive reasoning for the obtained results.

---

> ### Author Response · Authors · 2022-07-30
> **Re: Official Review of Paper5987 by Reviewer wokp**
>
> Thanks for the thoughtful review!
>
> Due to space limitations, we have to move many details on experiments to the Appendix. We will try our best
> to make the section on empirical results more useful and add a discussion/conclusion section at the end in the final version.

---

### Official Review · Reviewer_2T3t · 2022-07-11

**Rating:** 6
**Confidence:** 3
**Soundness:** 2 fair
**Presentation:** 3 good
**Contribution:** 2 fair

**Summary:**

The paper presents a new method for robust reinforcement learning that optimizes the dynamic and static CVaR. First the paper proves that the current action selection rule in distributional RL does not converge to the CVaR objective. The paper presents a new method that they show learns to optimiz static and dynamic CVaR. Experiments show that the proposed approach does better at the task of option trading than risk neutral and markov action selection startegies

**Questions:**

- What is meant by distributional RL does not converge to static or dynamic RL
- What about other literature on robust policy optimization? The paper can compare their method to these existing approaches
    -  Risk-constrained reinforcement learning with percentile risk criteria
    - Risk-sensitive and robust decision-making: a CVaR optimization approach
    - Tractable objectives for robust policy optimization


**Ethics Review Area:**

["I don’t know"]

**Limitations:**

I feel the paper currently lacks a good contrast to existing methods for robust RL, principally and experimentally.

**Strengths And Weaknesses:**

+ Interesting and relevant problem
+ Interesting theoretical and experimental results
- It is difficult to see how the method compares or positions itself to other papers in the robust RL literature
- The evaluation seem to be missing some relevant baselines as the paper points out itself that there is related work that can tackle static and dynamci CVaR

---

> ### Author Response · Authors · 2022-07-30
> **Re: Official Review of Paper5987 by Reviewer 2T3t**
>
> Thanks for the review!
>
> Question 1: Convergence of distributional RL
>
> As we pointed out in Section 2.1, there exist two commonly used definitions of CVaR for MDPs (static vs dynamic).
> We next show in Proposition 1 that the distributional-RL operator that is used in many existing works does not converge to either of this two versions of CVaR.
>
> Question 2: Comparison with other methods
>
> There are indeed many existing works in the literature that optimize for CVaR or other risk-sensitive objectives (including the three
> you listed). Many existing works are not based on RL -- they require a known reward/transition model. Some RL-based works are based on policy-gradient -- which are difficult to compare in a fair manner, as we explain in Section 4. In this work, we focus on approaches based on distributional RL, and compare with the most prominent baselines for distributional RL (QR-DQN and IQN).

---

> > ### Comment · Reviewer_2T3t · 2022-08-07
> > **response**
> >
> > I think it would be useful if the paper can position itself better in the existing literature. What is exactly that this method achieves that the current methods cannot (irrespective of focussing on distributional RL)? The current experiment domains do not help me either. How do the existing methods for CVaR maximization perform in these domains? Which of the three works that were mentioned are not based on RL? Why?

---

> > > ### Author Response · Authors · 2022-08-07
> > > **re: response**
> > >
> > > Thank you for your further questions!
> > >
> > > __What is exactly that this method achieves that the current methods cannot (irrespective of focussing on distributional RL)?__
> > >
> > > We address the problem of finding a CVaR-optimal policy with distributional RL -- which belongs to the family of "value-based" RL.
> > > The approaches of distributional RL have been demonstrated on various challenging domains (e.g. Atari games) in terms of the expected return. This work points out a problem that arises in existing methods that extend distributional RL to CVaR, in terms of its theoretical convergence. We propose a method to correct this. To the best of our knowledge, this is the first work that analyzes this issue and offers a solution that has better theoretical properties.
> > >
> > > __Can we solve for CVaR-optimal policies without using distributional RL?__ Of course. But we believe that distributional RL provides a path towards risk-sensitive policies that enjoys benefits including its simplicity (as in the case of Deep Q-Learning) and its well-tested, publicly available implementations. This motivates us to look closer at its theoretical aspects, resulting in the work we present in this paper.
> > >
> > > On the three works mentioned:
> > > 1) (Chow et al) Risk-constrained reinforcement learning with percentile risk criteria
> > >
> > > This work is based on RL. However, it is based on policy-gradient/actor-critic, which belongs to a different family of RL. As we pointed out in the paper, it is difficult to compare value-based RL with policy search methods in a fair manner. The method proposed by Chow et al is very sophisticated, but difficult to implement in practice, with no publicly available implementation (that we are aware of).
> > >
> > > 2) (Chow et al) Risk-sensitive and robust decision-making: a CVaR optimization approach
> > >
> > > This work is not based on RL. It assumes a known model and proposes a value-iteration (with linear interpolation) approach to solve CVaR MDPs.
> > >
> > > 3) (Chen & Bowling) Tractable objectives for robust policy optimization
> > >
> > > This work is not based on RL. It assumes that a distribution over the MDP parameters is given, and takes the Bayesian approach for robust MDPs. The algorithm involves solving a two-player zero-sum game using a form of counterfactual regret minimization.
> > >
> > > We hope that our explanations above address your concerns. Please do let us know if there are any other concerns regarding our work.

---

### Official Review · Reviewer_GmTT · 2022-07-11

**Rating:** 5
**Confidence:** 3
**Soundness:** 2 fair
**Presentation:** 3 good
**Contribution:** 3 good

**Summary:**

This paper considers learning a risk-sensitive policy based on CVaR via distributional RL. A modified distributional optimal operator is proposed with fixed-point satisfying CVaR optimality and an algorithm is proposed which combines QRDQN and the proposed operator. Experiments are executed on both synthetic and real-world data to present the performance of the proposed algorithm.

**Questions:**

1. Although the operator $\tilde{\mathcal{T}}_{\psi}$ defined in (10) may not result in convergence of distribution, I was wondering if $C_{\alpha}(\tilde{\mathcal{T}}_{\psi} U)$ converges to $C_{\alpha}(Z^{\pi^*})$ for all $U$.
2. The authors compare their proposed algorithm, which is based on QRDQN (Dabney et al. 2018b) with IQN(Dabney et al. 2018a). I think it would be more fair to use the same baseline structure to approximate the distribution function, either IQN or QRDQN.
3. The experiments on Atari games show that the proposed algorithm achieves a more concentrated distribution. Is there any benefit or effect with the concentration?

**Limitations:**

This paper requires that the rewards are drawn from a countable set and this is not a common assumption.

**Strengths And Weaknesses:**

Strengths:
This paper is well written and easy to follow. It indicates that the standard distributional Bellman operator does not work for the CVaR optimal policy and makes a proper modification.

Weakness:
The theoretical results are not solid enough to guarantee the convergence of the algorithm.

---

> ### Author Response · Authors · 2022-07-30
> **Re: Official Review of Paper5987 by Reviewer GmTT**
>
> Thanks for the thoughtful review!
>
> Question 1: On convergence of C_\alpha
>
> One is inclined to believe that this is true. Unfortunately we do not have a proof. We hope that this work can inspire further works in this direction that settle this question.
>
> Question 2: Comparison with QR-DQN and IQN
>
> We actually have results that compare with both QR-DQN and IQN using the same architecture for a fair comparison.
> For all experiments except the Atari games, we use the same QR-DQN architecture. For Atari games, we use the same IQN architecture.
> We will make this clearer in the Appendix.
>
> Question 3: On more concentrated distributions
>
> The more concentrated distributions suggest a more consistent behavior from play to play. When watching the playback video, we did observe that the strategy (subject to randomness of the game) is more consistent in each replay of the game.

---

### Meta-Review · Area_Chair_QmHs · 2022-08-27

**Recommendation:** Accept
**Confidence:** Less certain

**Metareview:**

This paper proposes a new action selection approach for risk-averse distributional reinforcement learning optimizing CVaR. It first shows that the action selection schemes used in existing approaches do not converge to the desired policies and subsequently shows that the fixed-point of the Bellman operator with the new action selection scheme is the desired optimal CVaR policy as long as it is stationary. It finally provides empirical results showcasing the benefits of the proposed approach.

The reviewers had mixed initial views on this paper. On the positive side, they found the paper to be well written and appreciated the new insights into the convergence of the existing action selection scheme as well as the more principled proposed scheme. On the negative side, there were concerns that (1) the paper does not actually convergence of the algorithm, only a fixed point, (2) that the paper does not provide sufficient discussion of the implications of the presented results, e.g. in the form of a conclusion and (3) that a comparison to CVaR optimization approaches that are not based on distributional RL is missing.
The authors' response addressed serval of these concerns so that all reviewers view this paper positively. Although, this paper still remains borderline and some concerns remain, the AC concurs with the reviewers that this paper has sufficient merits to be accepted, hence a recommendation for acceptance.

**Award:**

No

---

### Decision · Program_Chairs · 2022-09-14

Accept